# *Astragalus* Shiitake—A Novel Functional Food with High Polysaccharide Content and Anti-Proliferative Activity in a Colorectal Carcinoma Cell Line

**DOI:** 10.3390/nu14112333

**Published:** 2022-06-02

**Authors:** Bunu Tamang, Qi Liang, Biju Balakrishnan, Su Peng, Wei Zhang

**Affiliations:** 1Advanced Marine Biomanufacturing Laboratory, Centre for Marine Bioproducts Development, College of Medicine and Public Health, Flinders University, Bedford Park, SA 5042, Australia; tama0054@flinders.edu.au (B.T.); qi.liang@flinders.edu.au (Q.L.); bbal015@aucklanduni.ac.nz (B.B.); su.peng@flinders.edu.au (S.P.); 2School of Basic Medical Science, Shanxi University of Traditional Chinese Medicine, Taiyuan 030071, China

**Keywords:** shiitake mushroom, *Astragalus membranaceus*, hengshan *Astragalus* shiitake (HAS), polysaccharides, biotransformation, anti-proliferative activity, colorectal carcinoma (HCT 116)

## Abstract

The chemical and nutritional constituents of mushrooms can alter significantly when grown on different substrates. Based on this fact, an approach was made to cultivate a new type of mushroom, Hengshan *Astragalus* Shiitake, by growing Shiitake mushrooms on beds supplemented with the roots of an edible herbal plant, *Astragalus membranaceus*. In this study, three green extraction techniques, including microwave-enzyme assisted (MEA), ultrasound-enzyme assisted (UEA) and microwave-ultrasound-enzyme assisted (MUEA) extractions, were used to compare both the yield and antiproliferative activity of the polysaccharide-rich extracts (PREs) from HAS in human colorectal carcinoma cells (HCT 116). Both HAS-A and HAS-B extracts contain significantly higher amounts of polysaccharides when compared to the control (Shiitake extract), regardless of the extraction methods. The PREs from HAS-B have significantly higher anti-proliferative activity in HCT 116 compared to the control when using the UEA extraction method. Our findings demonstrate that HAS-B can become a novel functional food with anti-proliferative activities and the optimization of UEA extraction would help to develop new active extract-based health products.

## 1. Introduction

*Lentinus edodes*, commonly known as Shiitake, is one of the most popular and edible mushrooms native to East Asian countries like China and Japan [1]. Shiitake has high nutritional value and contains therapeutically active polysaccharides that benefit the immune system [2,3] and provide promising anti-cancer proliferative effects [4,5,6]. Numerous studies have provided solid evidence for their efficient bioactivities against breast cancer [7,8,9], lung cancer [10,11,12] and colorectal cancer [6,13]. However, studies have also shown that nutritional composition and biological activity of the constituents in mushrooms vary significantly when grown on different substrates [14]. Research was conducted where the spent mushroom substrate was supplied along with other nutrients for the cultivation of five different mushroom strains, including Shiitake [15]. This led to enhanced concentration of the polysaccharides and biomass of the mushrooms [15]. Similar effects were observed in the research conducted by Silwulski et al., where cultivation of mushrooms in different substrates (with different chemical composition) directly affected both the yield and nutritional composition [16]. Moreover, the immunomodulatory effect of Shiitake polysaccharides was also reported to be enhanced by growing Shiitake on Selenium (Se) supplemented substrate beds, where Se was transformed from the substrate into the mushroom polysaccharides with superior anti-cancer effects [17].

Based on the above facts, our industry collaborator (Shanxi Yulongxiang Agricultural Development Co. Ltd., Datong, China) developed a novel mushroom by growing Shiitake in the substrate beds supplemented with the roots of *Astragalus membranaceus*, a well-known edible herbal medicine with anti-cancer and immunomodulatory effects [18,19,20,21,22], and named it as Hengshan *Astragalus* Shiitake (HAS). Two different varieties of HAS, HAS-5% (HAS-A, growing on substrate with 5% *A. membranaceus* supplementation) and HAS-20% (HAS-B, growing on substrate with 20% *A. membranaceus* supplementation), were used in this study, which were proven to contain some bio-transformed components of *Astragalus*, determined by global metabolome analysis in our previous study [23]. According to the literature, one of the major active components providing anti-cancer effects in both *Astragalus* and Shiitake is polysaccharide [5,6,24,25]. Thus, the main objective of this study was to determine the changes in the extraction yield, chemical profile and anti-proliferative properties of the polysaccharide-rich extracts (PREs) from HAS in colorectal cancer cell line HCT-116 when compared to the PRE of Shiitake grown under the control condition. In addition, three different green extraction techniques were compared in this study for their extraction yield and bioactivity to understand the feasibility of developing health and nutritional products based the active HAS polysaccharide-rich extracts. 

The enzymatic extraction method is considered as one of the highly efficient, environmentally friendly and simple techniques for the extraction of polysaccharides from mushrooms (Cheng et al. 2015). As the fungal cell wall is mainly composed of cellulose, proteins, chitin, β-glucans and glycoproteins, the use of enzymes such as cellulases, proteases and pectinases help to degrade the cell wall and thus release the internal cellular components into the surrounding medium [26,27]. In addition, microwave and ultrasound assisted extractions have been proven as highly efficient techniques for the extraction of bioactives [28,29,30,31]. Microwaves assist the extraction process through the electro-magnetic waves that directly heat the extraction solvent rather than heat transfer from the vessel. Microwave heat allows minimal temperature gradient during the extraction [32]. On the other hand, ultrasound assists the extraction by forming a cavitation on the surface of materials [33,34] that allows the liquid bubbles near the solid surfaces to collapse vigorously, creating strong micro-jets to disrupt the cells.

## 2. Materials and Methods

### 2.1. Materials

The dried sample powders of *A. membranaceus*, Shiitake (*L. edodes*), HAS 5% (HAS-A) and HAS 20% (HAS-B) were provided by Shanxi Yulongxiang Agricultural Development Co. Ltd., China. The cultivation and preparation conditions were described in our earlier study [23]. The enzymes Alcalase and VinoTaste^®^ Pro were obtained as gift samples from Novozymes Pty Ltd., Sydney, Australia. Other enzymes and chemicals, including cellulase, pectinase, DEAE-Sephadex-CL 6B, dialysis membrane (molecular weight cut off 12,000 Da), glucose, Bio-Rad dye reagent based on Bradford assay and Bovine serum albumin (BSA), were purchased from Sigma Aldrich, Sydney, Australia.

The human colorectal carcinoma cells, HCT-116, were obtained from American Type Culture Collection (ATCC), and all other media and reagents including McCoy’s 5A medium, DMSO and MTT were purchased from Sigma Aldrich, Australia. The McCoy’s 5A media was supplemented with 10% fetal bovine serum (FBS), 1% penicillin (10,000 U/mL)—streptomycin (10,000 μg/mL) (Sigma Aldrich) and 1% glutamax (Thermo Fisher, Waltham, MA, USA). The apoptosis detection kit, Alexa Fluor 488 Annexin V/Apoptosis kit, was purchased from Invitrogen, Melbourne, Australia. All the reagents used in the experiments were of analytical or reagent grade.

### 2.2. Methods

#### 2.2.1. Extraction of Crude Polysaccharides

Extraction was carried out using three different extraction methods, including microwave-enzyme assisted (MEA), ultrasound-enzyme assisted (UEA) and microwave-ultrasound-enzyme assisted (MUEA) extraction methods. All extraction conditions were optimized according to previous studies. 

MEA extraction: The extraction was carried out based on the previously described method of [35], with some modifications. The mushroom powder sample (3 g) was mixed with 90 mL of distilled water (1:30 *w*/*v*) and vortexed to form a homogenous suspension. The sample–liquid mixture was then microwaved (Milestone start SYNTH microwave extractor) under the microwave power of 800 W for an initial 30 s to reach 60 °C, followed by 70 W for another 10 min, maintaining the same temperature [32]. Afterwards, the pH of the mixture was adjusted and maintained at 5.5 with 0.1 M citric acid and 0.2 M disodium hydrogen phosphate buffer [36]. Then, the mixture was hydrolysed at 50 °C for 20 min using an enzyme cocktail (Alcalase, Cellulase, Pectinase and VinoTaste^®^ Pro in the ratio of 1:1:1:1) with a concentration of 1% (*w*/*w*) of mushroom powder. This is a new enzyme cocktail developed for this study; most of the previous studies used only the three-enzyme cocktail of Alcalase, Cellulase and Pectinase in a ratio of 1:1:1. After 20 min of hydrolysis, the enzymes were deactivated by heating them in a water bath to a high temperature of 90 °C for 5 min. The mixture was then cooled and centrifuged at 5000× *g* for 10 min, and then the supernatant was collected. The supernatant was further precipitated with three volumes of absolute ethanol and stored at 4 °C overnight. The precipitates were then collected by centrifugation at 5000× *g* for 10 min and freeze-dried to obtain the PREs.

UEA extraction: The extraction was carried out based on the previously described methods of [33,34,37], with some modifications. The mushroom powder sample (3 g) was mixed with 90 mL of distilled water (1:30 *w*/*v*) and vortexed to form a homogenous suspension. The sample–liquid mixture was then exposed to pulsed ultrasonic power (750 Watt & 20 kHz Ultrasonic Processor) with 60% amplitude for 10 min. Then, the pH of the mixture was adjusted and maintained at 5.5 and hydrolyzed using the same four-enzyme cocktail under the same extraction condition and freeze-dried as described for the MEA extraction to produce the PREs.

MUEA extraction: The extraction was carried out based on the previously described method of [38], with some modifications. The mushroom powder sample (3 g) was mixed with 90 mL of distilled water (1:30 *w*/*v*) and vortexed to form homogenous suspension. The sample–liquid mixture was first extracted using ultrasound for 10 min, followed by microwave-assisted extraction for another 10 min using the same parameters described above. Eventually, this mixture was further enzymatically hydrolyzed and processed using the same process as the above experiments to produce the PREs.

#### 2.2.2. Characterization of the Polysaccharide-Rich Extracts

Total carbohydrate content: The total carbohydrate content (TCC) in each extract was determined using the phenol-sulfuric acid method, where D-Glucose was used as a reference [28]. The assay was conducted in a 96-well microplate, and 50 μL of sample/standard solution was taken for each analysis. To the above sample/standard, 30 μL of 5% Phenol solution was added, followed by rapid addition of 150 μL of 98% sulfuric acid. The reaction mixture was then heated in a water bath maintained at 90 °C for 5 min, and the absorbance was taken at 490 nm using a plate reader. Additionally, since the PREs were prepared by precipitation with alcohol, the majority of the precipitated carbohydrates would be polysaccharides, with only trace amounts of oligosaccharides and monosaccharides [32]. Hence, the TCC value was indicated as the total polysaccharide content and was calculated as below:$$\%\;\mathrm{Polysaccharide}=\frac{\mathrm{glucose}\;\mathrm{equivalent}\;\mathrm{concentration}\;\left(\frac{\mu g}{\mathrm{mL}}\right)\;\mathrm{in}\;\mathrm{test}\;\mathrm{sample}}{\mathrm{mushroom}\;\mathrm{powder}\;\left(\mathrm{sample}\right)\;\mathrm{concentration}\;\left(\frac{\mu g}{\mathrm{mL}}\right)\;\mathrm{used}}\times 100\%$$

Monosaccharides: The composition of monosaccharides in each sample was determined using HPLC with Phenomenex Kinetex C18 column (2.6 um 3 × 100 mm 100 A). Initially, the samples (10 mg) were hydrolysed with 100 μL of 72% sulfuric acid at 30 °C for 1 h. The concentration of sulfuric acid was diluted to 1 M and further incubated for 3–4 h at 100–110 °C for complete hydrolysis. After hydrolysis, the samples were spun at 16,000× *g* for 5 min to remove any insoluble components, then 20 μL of each sample and standards were transferred in a 1.5 mL Eppendorf tube. To each of the above tubes, 20 μL of 0.5 mM 2-deoxy Glucose was added as an internal standard. This was followed by the addition of phenyl methyl pyrazolone (PMP)/Ammonia mixture (0.5 M PMP and 1 M ammonium hydroxide solution), and the mixture was again incubated at 70 °C for 1 h. After incubation, 20 μL of 10 M formic acid was added to each tube, followed by 1 mL of di-butyl ether. This mixture was shaken vigorously for one minute, and later, the top di-butyl ether layer was removed carefully. The addition, vigorous mixing and removal of di-butyl ether were repeated twice-thrice until the reaction mixture was clear enough. Then, the above samples were transferred into a rotary evaporator for the removal of any traces of di-butyl ether. These samples were micro-centrifuged at 16,000× *g* for 5 min and then injected into the HPLC column for monosaccharide analysis [39,40].

Proteins: Protein content was estimated using the Bradford method. The assay was performed in a microplate, and bovine serum albumin (BSA) was used as a standard [41]. The Bio-Rad reagent was first diluted (1:4 dilutions in distilled water), and to every 5 μL of sample, 250 μL of diluted Bio-Rad reagent was added. The mixture was left to react for 15–45 min in the dark, and the absorbance was taken at 595 nm using a plate reader. The percentage of protein content was calculated as below:$$\%\;\mathrm{Proteins}=\frac{\mathrm{protein}\;\mathrm{equivalent}\;\mathrm{concentration}\;\left(\frac{\mu g}{\mathrm{mL}}\right)\;\mathrm{in}\;\mathrm{test}\;\mathrm{sample}}{\mathrm{mushroom}\;\mathrm{powder}\;\left(\mathrm{sample}\right)\;\mathrm{concentration}\;\left(\frac{\mu g}{\mathrm{mL}}\right)\;\mathrm{used}}\times 100\%$$

Molecular weight distribution: The molecular weight of polysaccharide extracts from each sample was determined by size exclusion chromatography using HPLC (PL aquagel-OH Mixed-H 8um 300 × 7.5 mm column). Dextran with a molecular weight ranging from 1–1100 kDa was taken as the reference standard, and 0.1 M sodium nitrate as a mobile phase. Initially, the samples were dissolved in MilliQ water at the concentration of 5 mg/mL and then micro-centrifuged at 13,000× *g* for 5 min. Then, 100 μL of the clear supernatant was mixed with 100 μL of 0.2 M sodium nitrate solution. This mixture was again spun at 13,000× *g* for 5 min, and 150 μL of this supernatant was injected into the HPLC column for molecular weight analysis [40].

#### 2.2.3. Anti-Proliferative Activities in HCT-116 Cells

The in vitro antiproliferative activity was determined in colorectal carcinoma cells, HCT-116. The cells were maintained in McCoy’s medium at 37 °C with 5% CO_2_ supplemented with 10% Fetal Bovine Serum, 1% penicillin/streptomycin and 1% Glutamax. Cells were seeded into a 96-well plate at a density of 5 × 10^3^ cells per well and incubated for 24 h [42]. These cells were then treated with each sample extract with the concentration of 200–3200 μg/mL. During treatment, 5-Fluorouracil, 1% DMSO in media and media alone were taken as the positive control, solvent control and the media control, respectively. The treated cells were incubated for 48 h at 37 °C with 5% CO_2_ supplemented. Subsequently, 100 μL of MTT (5 mg/mL diluted in media in a ratio of 1:10) was added to each well and again incubated for 4 h. The formazan crystals formed were then dissolved with 100 μL of DMSO and analyzed in the plate reader at 570 nm [25]. The percentage inhibition (i.e., anti-proliferative activity) was calculated as below:$$\mathrm{Percentage}\;\left(\%\right)\;\mathrm{inhibition}=\left(1-\frac{\mathrm{test}\;\mathrm{sample}\;\mathrm{absorbance}}{\mathrm{media}\;\mathrm{control}\;\mathrm{absorbance}}\right)\times 100\%$$

#### 2.2.4. Statistical Analysis

The data for each experiment were collected as three independent replicates and expressed as mean ± standard errors. Data were calculated and analyzed by Microsoft Excel 2016 and GraphPad Prism 8.0. The values with *p* < 0.05 were determined as statistically significant values and analyzed by one-way ANOVA followed by Dunnett’s and Tukey’s test.

## 3. Results

### 3.1. Effect of Different Extraction Methods on PRE Yield and Content of Total Polysaccharides

The total yield of PREs from three different extraction methods for four different samples is shown in Figure 1A. The total PRE yield from HAS-A and HAS-B were significantly higher than that of Shiitake (*p* < 0.01 and *p* < 0.0001, respectively), and the yield from HAS-B was significantly higher when compared to HAS-A (*p* > 0.001) in every extraction method. Thus, the yield increased in the following order, HAS-B > HAS-A > Shiitake > *A. membranaceus*. The highest yield was shown by the UEA extraction method, where the yield for HAS-B, HAS-A, Shiitake and *Astragalus* was 11.1 ± 0.29%, 8.07 ± 0.35%, 6 ± 0.13% and 2.5 ± 0.08 % *w*/*w*, respectively (Figure 1A). This trend was similar in the MEA and MUEA extraction methods, but there was no statistical difference in yield among the UEA, MEA and MUEA techniques. 

The total polysaccharide content from three different extraction methods for four different samples is shown in Figure 1B. The total polysaccharide contents from HAS-A and HAS-B were significantly higher (*p* < 0.05 and *p* < 0.01, respectively) when compared to Shiitake in the UEA and MUEA extraction methods. However, there was no significant difference in these values in MEA extracts and no statistical difference was observed between HAS-A and HAS-B in all the three extraction methods. The highest polysaccharide content in HAS-B, HAS-A, Shiitake and *Astragalus* was obtained by MUEA extraction, which was 66.2 ± 6.3%, 60.4 ± 1.6%, 48.5 ± 1.3% and 29.3 ± 4.05% *w*/*w*, respectively.

### 3.2. Effect of Different Extraction Methods on Monosaccharide Composition of Polysaccharide Extracts Analyzed by HPLC

The total monosaccharide content of the PREs from three different extraction methods for four different samples is shown in Figure 2, and the composition and concentration of monosaccharides are shown in Table 1. In all the three methods, the total monosaccharide content in HAS-A and HAS-B were significantly higher than that in the control Shiitake (Figure 2). In UEA extracts of HAS-B and HAS-A, the amounts of GlcAc and Gluc were significantly higher (GlcAc ~ 551.1 ± 1.5 μg/mL and Gluc ~ 2084.1 ± 1.5 μg/mL in HAS-B and GlcAc ~ 487.3 ± 1.8 and Gluc ~ 1876.8 ± 2.6 μg/mL in HAS-A) compared to the control Shiitake (GlcAc ~ 346.6 ± 0.8 μg/mL and Gluc ~ 1131.6 ± 2.3 μg/mL) (*p* < 0.0001). Meanwhile, the amount of Mannose in the same extract was significantly low, with 127.1 ± 1.5 in HAS-B (*p* < 0.001) and 160.3 ± 1.8 μg/mL in HAS-A (*p* < 0.05), when compared to the control Shiitake (171.3 ± 1.8 μg/mL). A similar pattern was observed in all the three extraction methods for the above three extracts (Table 1).

### 3.3. Effect of Different Extraction Methods on the Protein Content

The protein contents in the PRE from the three different extraction methods for four different samples are shown in Figure 3. The protein content in HAS-B extracts (MUEA and UEA) was significantly lower than the Shiitake (*p* < 0.001 and *p* < 0.05, respectively). However, there was no statistical difference in these values between HAS-A and HAS-B extracts in all the extraction methods. The lowest protein content for HAS-B, HAS-A, Shiitake and *Astragalus* extracts was obtained from the UEA extraction, which was 0.637 ± 0.21%, 1.294 ± 0.34%, 1.756 ± 0.31% and 1.535 ± 0.39% *w*/*w*, respectively. There was no significant difference in these values with MEA extracts.

### 3.4. Effect of Different Extraction Methods on Molecular Weight Profile of Polysaccharide-Rich Extracts

The molecular weight profiles of polysaccharides in PREs from three different extraction methods for four different samples are shown in Table 2. The relative molecular weight of HAS-B polysaccharides ranged from 1.2–822.8 kDa for MUEA extracts, 1.2–814 kDa for MEA extracts and 1.1–833 kDa for UEA extracts. These values were similar for HAS-A (Table 2), but for Shiitake polysaccharides, these values ranged from 1.2–10.5 kDa for MUEA, 1.2–10.4 kDa for MEA and 1.05–12.7 kDa for UEA extracts (Table 2). HAS-A and HAS-B contain higher molecular weight polysaccharides than Shiitake. When comparing HAS-A and HAS-B, the percentage concentration of higher molecular weight polysaccharides in HAS-A was significantly less (*p* < 0.01) than that in HAS-B. 

### 3.5. Effect of Different Extraction Methods on Anti-Proliferative Activity of PREs in HCT 116 Cells

The anti-proliferative activity of PREs from three different extraction methods for four different samples is shown in Figure 4 and Table 3. The MTT results showed dose-dependent antiproliferative activity of HAS-A and HAS-B in HCT 116 colorectal carcinoma cells after treatment for 48 h (Figure 4). The highest anti-proliferative activity for HAS-A, HAS-B, Shiitake and *A. membranaceus* was obtained by UEA extraction, where the extracts at 1.6 mg/mL caused 77 ± 2.82%, 94 ± 3.14%, 89 ± 2.24% and 60.35 ± 5.1% of cell death, respectively (Figure 4C). The IC_50_ value was 0.76 ± 0.122 mg/mL, 0.367 ± 0.044 mg/mL, 0.659 ± 0.059 mg/mL and 1.181 ± 0.145 mg/mL, respectively (Table 3). UEA extracts of HAS-B had the lowest IC_50_ value (*p* < 0.05) and higher anti-proliferative activity compared to the control Shiitake, while HAS-A had similar IC_50_ to the control (not statistically difference). On the other hand, the anti-proliferative activity of HAS-A, HAS-B and Shiitake with MEA and MUEA extraction methods was very low, with IC_50_ values more than three times higher than that of the UEA extraction method (Table 3). 

## 4. Discussion

In recent years, polysaccharides from biological origin are receiving greater attention in the medical field due to their strong therapeutic efficacy and effectiveness for a wide range of diseases such as cancer [4,6,43], cardiovascular diseases [44], diabetes [45,46], lung diseases [47] and kidney-associated diseases [48]. The polysaccharide components of both *Astragalus* and Shiitake have been proven to provide promising anti-proliferative properties in breast cancer [5,49], lung cancer [12,50] and colorectal cancers [6,51,52].

In this study, we investigated the yield, content and anti-proliferative activities of PREs (against colorectal cancer cells, HCT 116) from a new type of commercially produced Shiitake mushroom (HAS) using three green extraction methods to ensure the polysaccharide-rich extracts could be used for food and pharmaceutical industries. We found that the new approach of cultivating Shiitake in an edible medicinal plant *Astragalus*-supplemented bed helped in enhancing polysaccharide production in Shiitake mushrooms regardless of the extraction methods used. Taking the highest yield achieved by the UEA extraction method, the PRE yield for HAS-B and HAS-A increased by 85% and 35% when compared with the control Shiitake, respectively (Figure 1A). In contrast to our study, Yin et al. reported the highest yield, 9.38% *w*/*w* of Shiitake polysaccharides, using the MUEA extraction method [38], while we found only 6% *w*/*w* using MUEA in our study. The reason for lower yield in our study could be due to the modifications in previously reported extraction conditions. In the literature [35,38], the enzymatic hydrolysis was carried out for two hours before exposure to microwave or ultrasonic power. However, during our preliminary study, these conditions worked well only in terms of increasing the polysaccharide yield but were not promising in terms of providing higher anti-proliferative activity. For example, when the previously reported method of Yin et al. was used for UEA extraction of HAS-B [38], the yield was as high as 18.12% *w*/*w*, but its IC_50_ value against HCT 116 was ~2.56 mg/mL. Our modified UEA method had a comparatively lower yield of 11.1% *w*/*w* but had a significantly higher anti-proliferative activity in HCT 116 with IC_50_ value of 0.367 mg/mL. Hence, the modified condition in our experiment (exposing to microwave or ultrasonic power initially followed by enzyme hydrolysis for 20 min) was used with comparatively lower yield (Appendix A) but significantly improved anti-proliferative activity (*p* < 0.0001, in case of UEA HAS-B) compared to that of previously reported methods (Appendix A). Therefore, the enzymes were added in the sample/solvent mixture after exposing them to microwave/ultrasound in MEA, MUEA and UEA extractions in this study; these enzyme mixtures would have caused partial hydrolysis of the extracted polysaccharides, resulting in a decrease in PRE yield [35].

In addition, during our preliminary experiment, we compared the PRE yield and anti-proliferative effect of the extracts prepared from previously reported three-enzyme mixtures (Papain: Cellulase: Pectinase in a ratio 1:1:1) [27,38] with our new four-enzyme mixture (Alcalase: Cellulase; Pectinase: VinoTaste Pro^®^ at a ratio of 1:1:1:1). This test was only conducted in the UEA–HAS-B sample. There was no statistically significant difference in the PRE yield (Appendix A). But interestingly, the extracts prepared from our new four-enzyme mixture had a significantly higher anti-proliferative activity (*p* < 0.0001, for UEA HAS-B) when compared to the extracts prepared from previously reported three- enzyme mixture (Appendix A). For example, the IC_50_ value of the UEA–HAS-B extract using the previously reported three-enzyme mixture was 1.8 mg/mL, while the same sample extracted with our new four-enzyme mixture had a IC_50_ value of 0.367 mg/mL (Appendix A). Therefore, the new four-enzyme mixture was chosen for all the enzyme-assisted extractions in this study. The higher anti-proliferative activity could be due to the use of extra enzyme VinoTaste Pro^®^, the exo-(1,3) β glucanase, which would have cut the long (1,3) β glucan chain into a relatively shorter chain. The relatively smaller molecular weight polysaccharides can easily cross the biological membranes and show improved biological activity when compared to the larger ones [53,54]. In contrast to the polysaccharide content, the protein contents were lower in both HAS-A and HAS-B compared to the control Shiitake in each extraction method. Thus, it can be inferred that the cultivation of Shiitake in the beds supplemented with *A. membranaceus* activates the production of polysaccharides in the mushroom while inhibiting the protein synthesis. The molecular mechanisms involved in these processes can be studied in the future.

When the anti-proliferative effects of PREs from Shiitake, HAS-A and HAS-B were compared with the literature, the extracts from this study showed comparatively higher activity than that of published reports. For example, [27] reported nearly 71% cell death in HCT 116 and nearly 73% cell death in Hela cells at 1.8 mg/mL of Shiitake polysaccharides. Another experiment conducted by [55] showed the highest inhibitory action of Shiitake polysaccharides at 5 mg/mL, where the percentage inhibition of S-180, HCT 116 and HT 29 with their best polysaccharide fraction was ~97%, ~64% and ~53%, respectively. Furthermore, Unursaikhan et al. generated an O-sulfonated derivative of mushroom polysaccharide (lentinan) for better solubility and anti-proliferative activity; the modified derivative had the highest inhibition of ~68% at 5 mg/mL in S-180 cells [56]. The results from our studies showed a better anti-proliferative activity, i.e., ~89% and ~77% inhibitory effect in HCT 116 with 1.6 mg/mL of UEA-Shiitake and HAS-A polysaccharides, respectively. This inhibitory effect was more prominent with UEA-HAS-B polysaccharides, i.e., ~94% of inhibition at 1.6 mg/mL (Figure 4C). These results indicate that the extraction modifications made in our experiments are effective for extraction of bioactive polysaccharides from Shiitake, HAS-A and HAS-B. 

According to Zhang et al., the anti-proliferative activity of a polysaccharide molecule is directly related to its molecular weight and triple helix conformation [57]. Wang et al. further added that glucans with the molecular weights below 50 kDa cannot form a triple helix structure and hence cannot show prominent anti-proliferative activity [58]. Liu et al. also concluded that the glucans with a molecular weight between 290 and 2420 kDa exist as a triple helix conformation and those with a molecular weight of more than 1 × 10^6^ Da had better immune-enhancing properties [59]. However, our studies have shown significantly different results compared to the above-mentioned literature. In our study, the highest anti-proliferative activity of HAS-B, HAS-A and *A. membranaceus* in HCT 116 was obtained with the polysaccharide extracts of M_w_ ~ 1.1–832.7 kDa, ~1.2–976 kDa and ~2.5–113 kDa, respectively. Meanwhile, this molecular weight range was even lower in the case of Shiitake, where the highest inhibitory effect was shown by the polysaccharide extracts of M_w_ ~ 1.05–12.7 kDa (Table 3) which is completely different (extremely low) than what has been reported in the literature [58,60]. Therefore, in the future, determining the relationship between the molecular weight and bioactivities, including the anti-proliferative effects of these polysaccharides, would be critical for further optimization of extraction processes and bioactivities. Nonetheless, many studies have reported that high molecular weight polysaccharides showed better bio-activity only in vitro; when given in vivo, the immune system was highly triggered, and the molecules were engulfed by the phagocytes and macrophages and completely cleared off [61,62]. This suggests that the polysaccharide extracts with lower molecular weights in our study may have better anti-proliferative effects in vivo, which needs to be determined in future studies.

## 5. Conclusions

In conclusion, Shiitake mushrooms grown in the substrate beds supplemented with *Astragalus membranaceus* have a significantly higher amount of polysaccharides when compared to Shiitake grown under controlled conditions. The anti-proliferative activity of the PREs from HAS-A is not significantly different from that of the control Shiitake. However, the PREs from HAS-B have significantly higher anti-proliferative activity compared to the control Shiitake when optimum extraction conditions are used. Compared with MUEA and MEA extracts, UEA extracts had significantly higher anti-proliferative activity in HCT 116. In addition, the extracts from the UEA extraction method of the current study had significantly higher anti-proliferative activity in HCT 116 compared to previously reported studies. Hence, further investigation of the activities of HAS-B using the optimized extraction conditions and efficient purification would help to develop HAS as a promising functional food and/or a complementary medicine, particularly in the treatment and prevention of cancer.

## 6. Patents

The main research results from the work reported in this manuscript are involved in the filing of a patent; the authors will update this section once the process has been completed.

## Figures and Tables

**Figure 1 nutrients-14-02333-f001:**
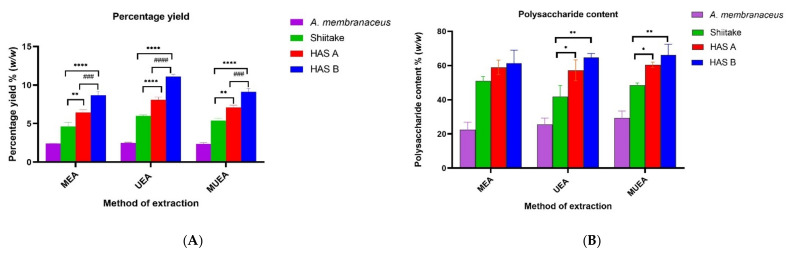
Effect of three different extraction methods on the PRE yield (**A**) and polysaccharide content (**B**) of *A. membranaceus*, Shiitake, HAS-A and HAS-B. The extraction methods are: MEA—Microwave/Enzyme assisted, UEA—Ultrasound/Enzyme assisted and MUEA—Microwave/Ultrasound/Enzyme assisted extractions. HAS-A and HAS-B significantly different from the control Shiitake with *p* < 0.05 indicated as *, *p* < 0.01 as **, and *p* < 0.0001 as ****, while ^#^ represents significant difference between HAS-A and HAS-B (*p* < 0.001 indicated as ^###^ and *p* < 0.0001 as ^####^).

**Figure 2 nutrients-14-02333-f002:**
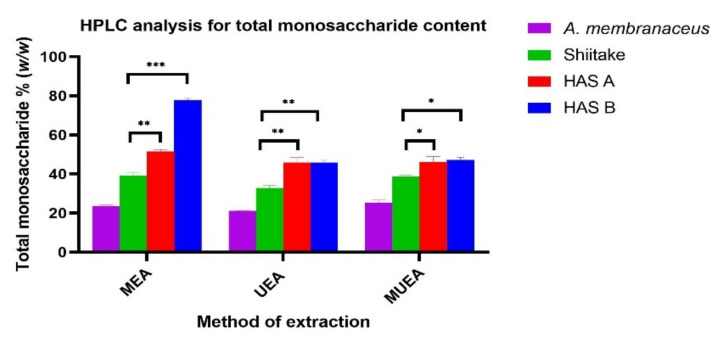
Effect of three different extraction methods on total monosaccharide content in the polysaccharide rich extracts of *A. membranaceus,* Shiitake, HAS-A and HAS-B, determined by HPLC. MEA—Microwave/Enzyme assisted, UEA—Ultrasound/Enzyme assisted, MUEA—Microwave/Ultrasound/Enzyme assisted extraction methods. HAS-A and HAS-B significantly different from the control Shiitake with *p* < 0.05 indicated as *, *p* < 0.01 as ** and *p* < 0.001 as ***.

**Figure 3 nutrients-14-02333-f003:**
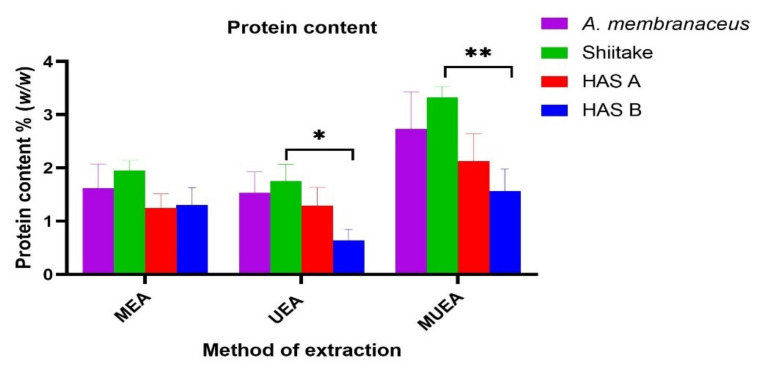
Effect of three different extraction methods on the protein content of *A. membranaceus*, Shiitake, HAS-A and HAS-B PREs. Protein content in HAS-B significantly different from control Shiitake with *p* < 0.05 indicated as * and *p* < 0.01 as **.

**Figure 4 nutrients-14-02333-f004:**
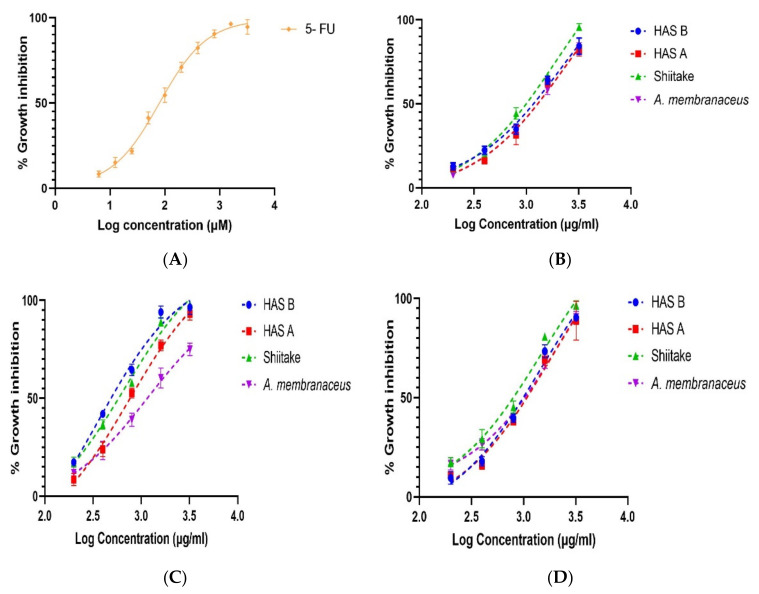
Effect of three different extraction methods on growth inhibition of HCT 116 after 48 h treatment with (200–3200) μg/mL of HAS-A, HAS-B, Shiitake and *A. membranaceus* PREs, determined by MTT assay (n = 3 with mean ± SD). (**A**) Growth inhibition by positive control, 5-FU. (**B**) Growth inhibition by MEA extracts. (**C**) Growth inhibition by UEA extracts. (**D**) Growth inhibition by MUEA extracts.

**Table 1 nutrients-14-02333-t001:** Composition and concentration of monosaccharides in MEA, UEA and MUEA extracts of *A. membranaceus*, Shiitake, HAS-A and HAS-B, determined by HPLC (concentration μg/mL ± S.D. n = 3).

Method of Extraction	Extracts	Monosaccharides Concentration μg/mL ± S.D									
		Man	Rib	Rha	GlcAc	GalAc	Gluc	Gal	Xyl	Ara	Fuc
MEA	*A. membranaceus*	60.0 ± 0.8	14.8 ± 0.2	43.6 ± 1.1	254.5 ± 7.1	61.1 ± 2	730.4 ± 15	303.6 ± 4.9	59.8 ± 1.1	209.8 ± 3	18.5 ± 0.3
	Shiitake	194.5 ± 6.3	13.0 ± 1.5	9.6 ± 0.8	392.8 ± 37	22.2 ± 2.4	1352.7 ± 47	785.1 ± 28.3	22.4 ± 0.5	3.2 ± 0.7	109.2 ± 4.1
	HAS-A	158.6 ± 4.2 ^##^	24.3 ± 1.8	11.8 ± 0.3	566.4 ± 23.9 ^#^	13.7 ± 9.6	2235.4 ± 70 ^###^	688.9 ± 22	17.9 ± 3.5	5.7 ± 0.2	97.6 ± 2.7
	HAS-B	174.6 ± 0.8 *	37.8 ± 3.6	11.1 ± 2.4	858.0 ± 6.6 ***	21.4 ± 12.1	3581.7 ± 27 ****	911.1 ± 2.2	17.1 ± 4.6	3.6 ± 0.3	156.0 ± 6
UEA	*A. membranaceus*	67.9 ± 0.6	8.5 ± 0.4	53.4 ± 1.2	165.1 ± 1	49.5 ± 0.2	634.8 ± 2.5	374.9 ± 6.7	30.9 ± 0.1	192.9 ± 3.2	19.0 ± 0.2
	Shiitake	171.3 ± 1.9	7.7 ± 0.6	8.9 ± 1.1	346.6 ± 0.9	25.1 ± 1.5	1131.7 ± 2.4	663.0 ± 36	24.4 ± 1.9	3.4 ± 0.3	90.8 ± 4.7
	HAS-A	160.3 ± 1.9 ^#^	9.5 ± 1.3	10.1 ± 1.4	487.3 ± 1.9 ^####^	21.1 ± 3.9	1876.9 ± 2.6 ^####^	736.6 ± 41.4	19.4 ± 0.7	4.7 ± 1.3	105.8 ± 6.2
	HAS-B	127.1 ± 1.5 ***	10.0 ± 0.1	13.9 ± 13.1	551.1 ± 1.5 ****	10.7 ± 1.2	2084.1 ± 1.5 ****	557.1 ± 63	12.8 ± 1.2	5.0 ± 0.6	100.7 ± 12
MUEA	*A. membranaceus*	53.7 ± 2.9	9.1 ± 1.6	44.8 ± 2.5	298.3 ± 25.9	45.1 ± 5	901.7 ± 37.7	313.1 ± 14.6	27.5 ± 0.48	169.0 ± 9.1	16.3 ± 0.5
	Shiitake	193.4 ± 3.8	9.7 ± 2.5	9.2 ± 0.8	385.7 ± 3.8	25.8 ± 0.2	1365.3 ± 23.8	749.2 ± 15.1	21.5 ± 0.46	3.4 ± 1.3	103.5 ± 2
	HAS-A	149.2 ± 7.3 ^#^	40.7 ± 5.5	7.0 ± 0.9	501.4 ± 43	19.0 ± 8.7	1892.1 ± 97 ^#^	697.0 ± 35.5	17.5 ± 2.2	3.9 ± 2.2	101.9 ± 5.8
	HAS-B	94.8 ± 9.4 **	29.2 ± 5.3	10.4 ± 0.2	533.1 ± 40.8 *	11.0 ± 2.5	2066.0 ± 61 *	512.9 ± 45.3	10.2 ± 0.21	3.5 ± 0.8	89.8 ± 8

Note: Man (mannose), Rib (ribose), Rha (rhamnose), GlcAc (glucuronic acid), GalAc (galacturonic acid), Gluc (glucose), Gal (galactose), Xyl (xylose), Ara (arabinose) and Fuc (fucose). HAS-A and HAS-B significantly different from control Shiitake indicated as # and *, respectively. *p* < 0.05 indicated as #/*, *p* < 0.01 as ##/**, *p* < 0.001 as ###/*** and *p* < 0.0001 as ####/****.

**Table 2 nutrients-14-02333-t002:** Relative molecular weight profiles of MEA, UEA and MUEA polysaccharide-rich extracts of HAS-B, HAS-A, Shiitake and *A. membranaceus* determined by size exclusion chromatography by HPLC.

Extracts	Major Peaks for HAS-B			Major Peaks for HAS-A			Major Peaks for Shiitake			Major Peaks for *Astragalus*		
	Rt, mins	% Concen-tration	M_w_, kDa	Rt, mins	% Concen-tration	M_w_, kDa	Rt, mins	% Concen-tration	M_w_, kDa	Rt, mins	% Concen-tration	M_w_, kDa
MEA	12.885	12.901 ***	813.967	12.64	0.031	1092.567	16.514	80.829	10.399	15.055	1.953	60.021
	16.015	33.56	18.94	16.693	72.565	8.387	17.605	10.873	2.804	16.665	56.546	8.674
	17.037	22.164	5.547	17.604	12.339	2.807	18.307	6.821	1.206	17.82	26.399	2.165
	17.655	10.848	2.64	18.315	7.68	1.195				18.318	10.229	1.19
	18.316	10.973	1.193									
UEA	12.866	12.983 **	832.763	12.734	0.194	975.885	16.35	20.508	12.72	14.629	4.399	100.135
	16.028	25.044	18.646	17.687	78.288	2.541	17.17	42.988	4.73	17.088	29.413	5.218
	17.719	36.966	2.445	18.326	16.625	1.179	17.5	22.04	3.19	17.706	39.937	2.483
	18.344	15.217	1.154				18.41	12.25	1.05	18.333	11.824	1.169
MUEA	12.876	20.234 ****	822.817	12.787	4.685 ^##^	915.679	11.375	75.276	10.5	15.304	1.029	44.501
	15.953	53.484	20.404	16.503	70.536	10.537	16.506	12.763	2.698	16.733	57.15	7.993
	17.638	8.581	2.695	17.604	9.554	2.807	17.637	10.199	1.213	17.671	25.882	2.59
	18.305	6.126	1.209	18.29	4.454	1.231	18.302			18.311	12.152	1.2

Note: The percentage concentration of higher molecular weight polysaccharides was significantly higher in HAS-B (*p* < 0.01 indicated as **, *p* < 0.001 as *** and *p* < 0.0001 as ****) and HAS-A (*p* < 0.01 indicated as ##) when compared to Shiitake.

**Table 3 nutrients-14-02333-t003:** Effect of three different extraction methods on IC_50_ values of HAS-A, HAS-B, Shiitake and *Astragalus* in HCT 116 determined by MTT assay.

Extracts	IC_50_ (mg/mL) ± S.D. (n = 3)			
	HAS-B	HAS-A	Shiitake	*A. membranaceus*
MUEA	1.444 ± 0.244	1.78 ± 0.443	1.491 ± 0.379	2.417 ± 0.298
MEA	2.224 ± 0.206	2.395 ± 0.52	1.975 ± 0.3	2.346 ± 0.181
UEA	0.367 ± 0.044 *	0.76 ± 0.122 ^##^	0.659 ± 0.059	1.181 ± 0.145

Note: * IC_50_ value for UEA-HAS-B extract was significantly low (*p* < 0.05 indicated as *) in HCT 116 compared to UEA Shiitake, while ## indicates IC_50_ value of HAS-A is significantly higher than HAS-B (*p* < 0.01).

## Data Availability

Data are available in a publicly accessible repository. The data presented in this study are openly available in FigShare at https://doi.org/10.6084/m9.figshare.19221252.v1 (accessed on 23 February 2022).

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
