# Peer review of "Astragalus Shiitake—A Novel Functional Food with High Polysaccharide Content and Anti-Proliferative Activity in a Colorectal Carcinoma Cell Line"

_nutrients, 2022, doi:10.3390/nu14112333_

Round 1

Reviewer 1 Report

Dear authors,

I have carefully read your manuscript and ask you to make several minor corrections:

line 44 - please disclose your industry collaborator for the fist time you mentioned it

line 68 - please add clarification that Bio-Rad dye reagent is based on Bradford assay

line 92 - please correct a typo in "alcalAse"

line 93-95 - as I found in google VinoTaste® Pro is a combination of pectinase and beta-gluconase, so what was the reason to add pectinase as a separate component rather than just use combination of alcalase, cellulase and VinoTaste® Pro?

line 161 - "in vitro" without dash

line 257 - kDa without space

Also, I ask you to clarify and add in the manuscript discussion around following topics:

  1. Please add short description of biochemical mechanism of enzymes action and its importance for extraction procedure.
  2. If Cellulase, Pectinase and beta-Gluconase hydrolyze polysaccharides to mono- and oligosaccharides how you discriminated polysaccharides from monosaccharides? Moreover phenol-sulfuric acid assay is able to measure both poly- and monosaccharides, so for the polysaccharide content measurement monosaccharides should be removed with dialysis or any other suitable method prior phenol-sulfuric assay.
  3. You observed increased monosaccharide content with MEA extraction in comparison with UEA and MUEA. Are there any mechanistical explanation why MEA is better than other methods? Why combination of Ultrasound and Microwave is worse than Microwave alone?
  4. While you did not separate saccharides from other molecules why you believe that biological activity of the extract was caused by saccharide component?
  5. MTT assay reflects metabolic activity, and it is non-direct method to evaluate toxicity or proliferation. From biological point of view cytotoxic and anti-proliferative action are different, please clarify did you mean that extracts are able to stop proliferation (due to cell cycle arrest) or cause cell death?
  6. While you use only one cell line and did not use any normal cells as control you should be very careful in statements regarding cancer treatment or prevention. Please add discussion about needs to evaluate mechanism of action and possible toxicity in normal (non-cancer) cells.

Author Response

Dear Reviewer, 

We appreciated your comments on this manuscript. All your comments have been addressed point by point.  Please see the attachment.

Thanks for your time. 

With the very best wishes

Reviewer 2 Report

The paper entitled "Astragalus Shiitake - a novel functional food with high polysaccharide content 
and anti-proliferative activity in a colorectal carcinoma cell line" by Tamang et al. reports study 
on activity of polysaccharide fraction of astragalus cultivated in the substrate supplemented with Astragalus membraceus. 
The study reports higher anti-proliferative activity in HCT 116 of this type of extracts. 

The study lacks determination of cytotoxicity in normal, healthy cells. 
The experiments were made in triplicate, this may be too little to reliably determine significant differences between samples. 

Minor remarks:

line 12-13 "A new mushroom .... was developed" - please change, it sounds misleading
line 13 "Astragalus membraceus" -> italic
line 14 "(MEA, UEA and MUEA extraction)" - please explain abbreviations
line 45, line 62 "supplemented with Astragalus membranaceus," - what part of the plant?

Author Response

(The authors gave the same response as above.)

Round 2

Reviewer 1 Report

Dear authors,

Thank you for the detailed answers. I do not have any questions and recommend to accept your manuscript.

Reviewer 2 Report

The questions were addressed and the manuscript was improved